



# NeverWorld2: An idealized model hierarchy to investigate ocean mesoscale eddies across resolutions

Gustavo M. Marques[1], Nora Loose[2], Elizabeth Yankovsky[3], Jacob M. Steinberg[4], Chiung-Yin Chang[5], Neeraja Bhamidipati[5], Alistair Adcroft[5], Baylor Fox-Kemper[7], Stephen M. Griffies[5,6], Robert W. Hallberg[5,6], Malte F. Jansen[8], Hemant Khatri[9], and Laure Zanna[3]

[1]Climate and Global Dynamics Laboratory, National Center for Atmospheric Research, Boulder, CO
[2]University of Colorado, Department of Applied Mathematics, Boulder, CO
[3]Courant Institute, New York University, New York, NY
[4]Woods Hole Oceanographic Institution, Woods Hole, MA
[5]Atmospheric and Oceanic Sciences, Princeton University, Princeton, NJ
[6]NOAA Geophysical Fluid Dynamics Laboratory, Princeton, NJ
[7]Department of Earth, Environmental, and Planetary Sciences, Brown University, Providence, RI
[8]Department of the Geophysical Sciences, The University of Chicago, Chicago, IL
[9]Department of Earth, Ocean and Ecological Sciences, University of Liverpool, Liverpool, UK

**Correspondence:** Gustavo M. Marques (gmarques@ucar.edu)

**Abstract.** We describe an idealized primitive equation model for studying mesoscale turbulence and leverage a hierarchy of grid resolutions to make eddy-resolving calculations on the finest grids more affordable. The model has intermediate complexity, incorporating basin-scale geometry with idealized Atlantic and Southern oceans, and with non-uniform ocean depth to allow for mesoscale eddy interactions with topography. The model is perfectly adiabatic and spans the equator, and thus fills a

gap between quasi-geostrophic models, which cannot span two hemispheres, and idealized general circulation models, which generally have diabatic processes and buoyancy forcing. We show that the model solution is approaching convergence in mean kinetic energy for the ocean mesoscale processes of interest, and has a rich range of dynamics with circulation features that emerge only due to resolving mesoscale turbulence.

## 1   Introduction

Mesoscale eddies have a profound impact on the transport of properties in the ocean. They affect the currents, stratification, ocean dynamic sea level variability, and uptake of physical and biogeochemical tracers. Eddies thus play an important role in regulating climate on regional and global scales and on timescales of weeks to centuries. Mesoscale eddies form on spatial scales near the baroclinic Rossby deformation radius (Smith and Vallis, 2002; Arbic and Flierl, 2004; Thompson and Young, 2007; Hallberg, 2013). The deformation radius varies regionally between 10-100 km horizontally (Chelton et al., 1998). These

scales are too small to be resolved globally in routinely used structured-grid ocean climate simulations and, therefore, must be parameterized.

The most notable scheme of mesoscale eddy transport implemented in climate models is the Gent McWilliams (GM) parameterization, which mimics the effects of baroclinic instability by flattening isopycnals and acting as a net sink of available





potential energy (Gent and McWilliams, 1990; Gent et al., 1995). Motivated by the effect on available potential energy, eddy
kinetic energy parameterizations have been developed (Cessi, 2008; Eden and Greatbatch, 2008; Marshall and Adcroft, 2010),
to keep track of the mechanical energy in idealized simulations (Marshall et al., 2012) and/or scale the GM parameter in
global climate simulations (Adcroft et al., 2019). The specific goal of this paper is to present a test for such energy-based
parameterizations, although mesoscale parameterizations based on other approaches can also be tested in the same framework.

As the horizontal grid spacing of climate models is refined, such that the grid box size becomes comparable to the defor-
mation scale, a regime commonly referred to as the "grey zone" is reached. In this regime, some eddies are being partially
resolved but the resolution does not allow for their effects on the large scale current and stratification to be fully accounted for
(Hallberg, 2013). In particular, the inverse kinetic energy cascade (or backscatter) and the barotrozipation of the flow remain
too weak in both idealized (Jansen and Held, 2014) and global models (Kjellsson and Zanna, 2017). A large meridional extent
and continental slopes elicit a grey zone at some latitudes and depths under virtually all stratifications in eddying simulations,
although their extent is controlled by model resolution. Recent mesoscale parameterizations focus on these two aspects with
novel momentum closures (Bachman, 2019; Jansen et al., 2019).

The majority of these mesoscale parameterizations have been developed independently, using different dynamical assump-
tions (e.g., quasi-geostrophic dynamics or primitive equations) and different idealized configurations with limited spatial extent
(e.g., double gyre or channel). This approach has led to a lack of coherent and robust analysis on the effect of eddies and their
parameterizations on the ocean dynamics.

Here, we present an idealized model to capture the essence of mesoscale eddy dynamics at varying horizontal grid resolu-
tions, investigate the effect of mesoscale eddies on the large-scale dynamics, and serve as a framework for testing and evaluating
eddy parameterizations. The model allows for a clean and extensive analysis of the dynamics and energetics of the flow as a
function of horizontal resolution, which is often limited in primitive equation and diabatic global models due to computational
resources (Hewitt et al., 2020; McClean et al., 2011).

We introduce a model configuration - referred to as NeverWorld 2 (NW2), which is an extension of the Southern Hemisphere-
only NeverWorld configuration presented in Khani et al. (2019) and Jansen et al. (2019). NW2 is a stacked shallow water model
configuration with idealized geometry comprising a single cross-equatorial basin and a re-entrant channel in the Southern
Hemisphere. The broad configuration is similar to that of Wolfe and Cessi (2009), except NW2 is strictly adiabatic, on a
spherical grid, and forced only by winds. The global volume of water in each density layer is set by the initial conditions,
with the dynamics determining the spatial distribution of stratification which can adjust locally. A hierarchy of horizontal
grid resolutions allows us to consider mesoscale eddies in distinct dynamical regimes, e.g., Southern Ocean-like dynamics,
mid-latitude gyres and equatorial flows. This hierarchy also encompasses coarse (unresolved), grey (partially resolved) and
mesoscale eddying (fully resolved) flow regimes in portions of the domain controlled by the selected model resolution.
We discuss the model equations, the convergence of the simulations as a function of resolution, and the energetics of the
flow. This paper is meant to be an introduction to the configuration and the datasets for use by the community to understand,
test and evaluate mesoscale dynamics and novel closures.



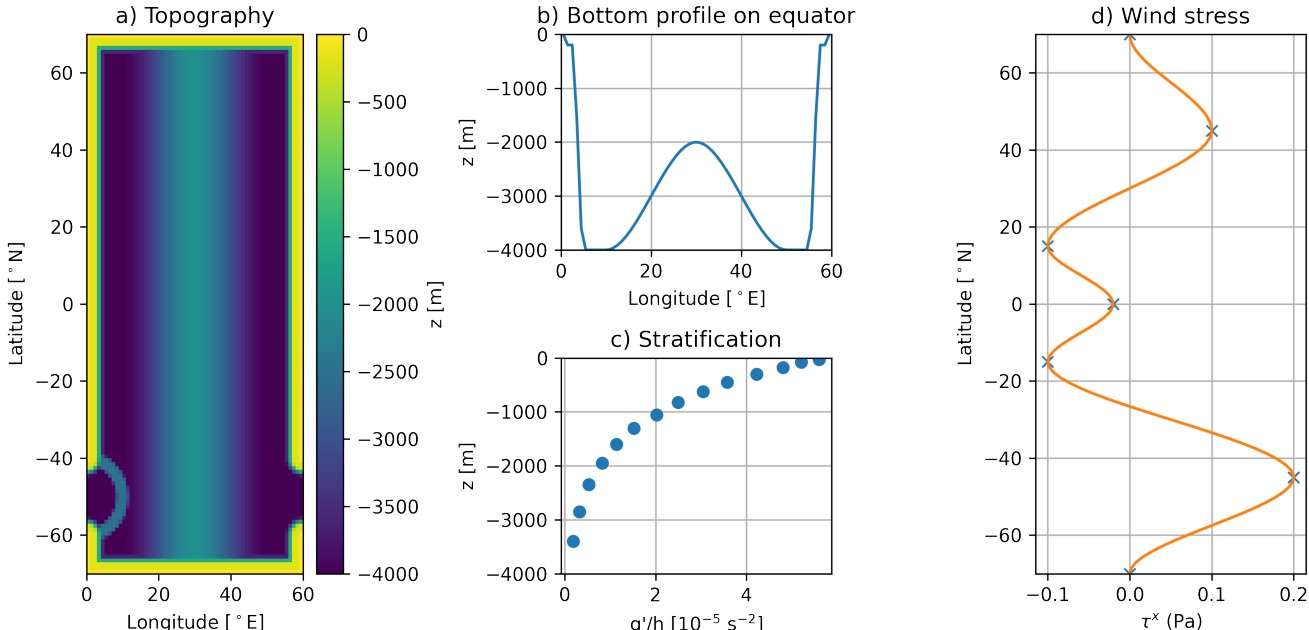

**Figure 1.** NeverWorld2: a) Bathymetry (depth in [m]); b) Cross-section of bathymetry on the equator; c) Stratification, $2g'_{k+1/2}/(h_k+h_{k+1})$, at initial depths of each interface; d) Latitudinal profile of zonal wind stress forcing (in Pa), constructed from piecewise cubics with nodes indicated by crosses.

## 2  Model Description

### 2.1  Model Equations

We consider an adiabatic and hydrostatic fluid system with a single thermodynamic constituent, simplified to a linear equation of state. This system can be approximated by $N$ stacked layers of piecewise constant density. The equations of motion for the primary prognostic model variables, which are the zonal flow ($u$), meridional flow ($v$), and layer thickness ($h$), are written in vector-invariant form:

$$\partial_t u_k - q_k v_k h_k + \partial_x K_k + \partial_x M_k = \mathcal{F}_k^x, \tag{1}$$

$$\partial_t v_k + q_k u_k h_k + \partial_y K_k + \partial_y M_k = \mathcal{F}_k^y, \tag{2}$$

$$\partial_t h_k + \partial_x (u_k h_k) + \partial_y (v_k h_k) = 0, \tag{3}$$

with the vertical stress divergence and horizontal friction given by

$$\mathcal{F}_k^x = \frac{1}{\rho_o h_k}\left(\tau_{k-1/2}^x - \tau_{k+1/2}^x\right) - \nabla \cdot \nu_4 \nabla\left(\nabla^2 u_k\right) \tag{4}$$

$$\mathcal{F}_k^y = \frac{1}{\rho_o h_k}\left(\tau_{k-1/2}^y - \tau_{k+1/2}^y\right) - \nabla \cdot \nu_4 \nabla\left(\nabla^2 v_k\right), \tag{5}$$




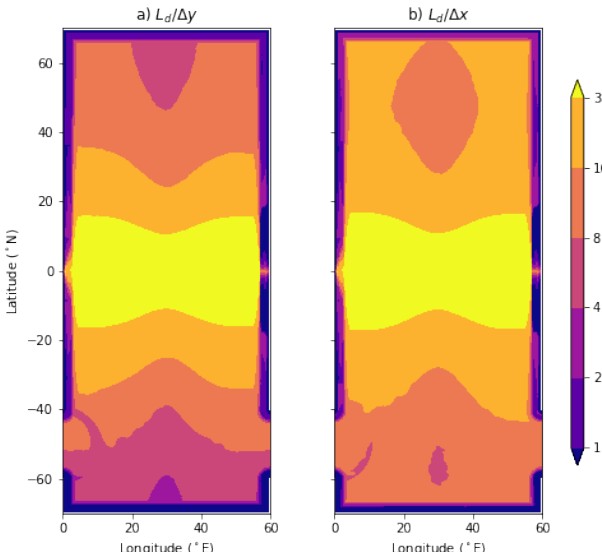

**Figure 2.** a) Meridional and b) zonal resolution parameters, for the $1/32°$ configuration, shown as the deformation radius $L_d$ divided by the model meridional $\Delta y$ or zonal $\Delta x$ extent, respectively. The deformation scale is computed from the spun-up state. The aspect ratio between meridional and zonal extent is close to unity for most of the domain except for the very highest latitudes of the southern ocean where the deformation radius is least well resolved.

where a subscript $k$ indicates the vertical layer number with $k = 1$ the top-most, and $k = N$ the bottom-most. We use the short-hand $\partial_t$, $\partial_x$, and $\partial_y$, for partial derivatives in time, zonal, and meridional directions, respectively. $\nabla\cdot$ is the horizontal divergence, $\nabla$ the horizontal gradient, and $\nabla^2 = \nabla \cdot \nabla$ is the horizontal Laplacian. The MOM6 code (Adcroft et al., 2019) that discretizes these equations makes use of horizontal orthogonal curvilinear coordinates, though we here write the more concise Cartesian coordinate notation for brevity.

Other dynamic quantities that are derived from the primary variables include:

- the interface positions, $\eta_{k-1/2} = -D + \sum_{l=k}^{N} h_l$, indicated with half-integer labels;

- the potential vorticity, $q_k = \frac{1}{h_k}\left(f + \partial_x v_k - \partial_y u_k\right)$;

- the kinetic energy per mass, $K_k = \frac{1}{2}\left(u_k^2 + v_k^2\right)$;

- the Montgomery potential, $M_k = \sum_{l=1}^{k} g'_{l-1/2}\eta_{l-1/2}$;

- the dynamic lateral viscosity,

$$\nu_4 = C_4 \frac{\Delta^4}{8\pi^2}\sqrt{\left(\partial_x u_k - \partial_y v_k\right)^2 + \left(\partial_y u_k + \partial_x v_k\right)^2},$$





where $\Delta^4$ is the fourth power of the grid-spacing which follows a particular discretization, as proposed in the Appendix of Griffies and Hallberg (2000) and is different than simply using the square of the cell area;

– the vertical stress, $\boldsymbol{\tau}_{k-1/2} = -A_v \frac{\rho_o}{h_{k-1/2}}\left(\boldsymbol{u}_{k-1} - \boldsymbol{u}_k\right)$;

– the bottom stress, $\boldsymbol{\tau}_{N+1/2} = -C_d \rho_o |\boldsymbol{u}_B| \boldsymbol{u}_N$, which uses a quadratic drag law and where $\boldsymbol{u}_B$ is the flow averaged over the bottom-most 10 m.

The surface wind stress, $\boldsymbol{\tau}_{1/2}$, is prescribed, fixed in time, and is distributed over the top 5 m. The remaining parameters are the reduced gravity of each layer, $g'_{k-1/2}$, a reference density, $\rho_o = 1000$ kg m$^{-3}$, the Coriolis parameter, $f = 2\Omega\sin\phi$ (with $\Omega = 7.2921 \times 10^{-5}$ s$^{-1}$ and $\phi$ is latitude), the background kinematic vertical viscosity, $A_v = 1.0 \times 10^{-4}$ m$^2$ s$^{-1}$, a dimensionless bottom drag coefficient, $C_d = 0.003$, and the bottom depth, $z = -D(x, y)$. We have chosen to use a biharmonic dissipation operator with a dimensionless Smagorinsky coefficient of $C_4 = 0.2$, which is larger than the recommended range

suggested by Griffies and Hallberg (2000). This is to ensure sufficient dissipation in the absence of any other parameterizations of lateral friction. Other scale-aware alternatives (Bachman et al., 2017) arrived at comparable results.

We provide analysis of the energetics here, and in subsequent papers. To facilitate such analysis, it is useful to write out the energy budget equations. The kinetic energy (KE) in layer $k$ is given by $\mathrm{KE}_k = h_h K_k$. To obtain the KE equation, we add $u_k h_k \times (1)$, $v_k h_k \times (2)$ and $K_k \times (3)$, which gives


$$\partial_t \left(K_k h_k\right) + \nabla \cdot \left(K_k \boldsymbol{u}_k h_k\right) + \boldsymbol{u}_k h_k \cdot \nabla M_k = \boldsymbol{u}_k h_k \cdot \boldsymbol{\mathcal{F}}_k \,. \quad (6)$$

The Coriolis term does not contribute to equation (6). However, the numerical model uses a C-grid staggering of variables so that locally the numerical Coriolis terms do not drop out, thus affecting KE. We use the Arakawa and Hsu (1990) discretization of Coriolis terms which, when integrated over the whole domain, conserves total KE for horizontally non-divergent flow. The

term $\boldsymbol{u}_k h_k \cdot \nabla M_k$ is the conversion between potential energy and KE.

The potential energy (PE) at interface $k - \frac{1}{2}$ is given by $\mathrm{PE}_{k-1/2} = \frac{1}{2} g'_{k-1/2} \eta_{k-1/2}^2$. The corresponding PE equation is obtained by vertically summing equation (3) from the bottom up to interface $k - \frac{1}{2}$ and then multiplying by $g'_{k-1/2} \eta_{k-1/2}$:

$$\partial_t \left(\frac{1}{2} g'_{k-1/2} \eta_{k-1/2}^2\right) + \nabla \cdot \left(g'_{k-1/2} \eta_{k-1/2} \sum_{l=k}^{N} \boldsymbol{u}_l h_l\right)$$


$$= g'_{k-1/2} \left(\sum_{l=k}^{N} \boldsymbol{u}_l h_l\right) \cdot \nabla \eta_{k-1/2} \,. \quad (7)$$

When summed in the vertical, the right-hand side of equation (7) and the PE conversion term in equation (6) are equal.

The available potential energy (APE) is the domain integrated PE minus the domain integrated PE of the resting state. The resting state, and interface positions used at initialization, are given by $\eta_{k-1/2}(t = 0) = \max\left(z_{k-1/2}^0, -D\right)$, where $z_{k-1/2}^0$ is a constant nominal position for each interface. In this adiabatic model, changes in the domain integrated PE are exactly the

changes in APE, with no approximations or ambiguity.





## 2.2 Configuration

The NW2 configuration is set up as follows. The domain is a sector of a sphere with angular width $60°$, and with a single basin and a re-entrant channel in the Southern Hemisphere. The basin is bounded by solid coasts at $±70°$N/S. Not extending to the poles avoids infinitesimal spherical coordinate cells as the meridians converge. We use a regular spherical grid rather than a Mercator grid so that the placement of boundaries (extents of the grid) is exactly the same for all resolutions. The regular spherical grid means the cells are distorted with cell-wise average aspect ratio $\Delta y/\Delta x \sim 1.4$, exceeding 2 only poleward of $±60°$. The full bathymetry is shown in Fig. 1a, with a cross-section along the equator in Fig. 1b. The depth of the continental shelf is 200 m and it has a nominal width of $2.5°$. A cubic profile, of width $1/8^{\text{th}}$ of the shelf, connects the shelf to the beach (which has zero depth out to $1/8^{\text{th}}$ of the shelf width). Another cubic profile of width $2.5°$ connects the shelf to the abyss with nominal depth of 4000 m. A large abyssal ridge of height 2000 m runs north-south down the middle of the basin with a cubic profile and radius of $20°$. The re-entrant channel spans $60°$S-$40°$S and a semi-circular ridge of height 2000 m, radius of $10°$, and thickness of $2°$ is centered on the channel opening in the west to block the deep flow through the channel. These deep ridges are idealizations of the mid-Atlantic ridge, and the Scotia arc that acts as a sill across the Drake passage.

The wind-stress is strictly zonal and fixed in time (Fig. 1b), and is an idealization of the mean zonal wind profile (see for example Fig. A1 of Chaudhuri et al., 2013). We construct the wind stress from piecewise cubic functions that interpolate between the values 0, 0.2, $-0.1$, $-0.02$, $-0.1$, 0.1 and 0 Pascals at latitudes $-70°$, $-45°$, $-15°$, $0°$, $15°$, $45°$ and $70°$, respectively. Each interpolation node has zero derivative so that both the wind stress and the curl of wind stress are zero at the north and south boundaries.

The initial stratification and vertical resolution are intimately linked (Fig. 1c). We use 15 layers and choose nominal thicknesses (meters) of 25, 50, 100, 125, 150, 175, 200, 225, 250, 300, 350, 400, 500, 550, 600. The adiabatic conditions relieve the model of resolving surface boundary layer processes, but finer resolution near the surface is preserved to accurately capture surface intensification of mesoscale energy (Smith and Vallis, 2002). Actual initial thicknesses are the shallower of this nominal profile and whatever is clipped by topography to yield flat interfaces in the interior. The reduced gravity at each interface has values (m s$^{-2}$) 10, 0.0021, 0.0039, 0.0054, 0.0058, 0.0058, 0.0057, 0.0053, 0.0048, 0.0042, 0.0037, 0.0031, 0.0024, 0.0017, 0.0011 (m s$^{-2}$). The first value corresponds to the gravitational acceleration at the surface. The implied density profile is approximately exponential with a depth scale of 1000 m (approximate due to rounding input parameter values to 2 significant digits).

The ratio between the first baroclinic deformation radius ($L_D$) to the meridional and zonal grid spacing, is shown in Fig. 2a and Fig. 2b, respectively. Assuming that to resolve eddies, $L_D/\Delta x \geq 2$ (Hallberg, 2013), Fig. 2 highlights that the $1/32°$ horizontal resolution simulation explicitly resolves mesoscale eddies, except on the shelves and at very high latitudes of the Southern Ocean.





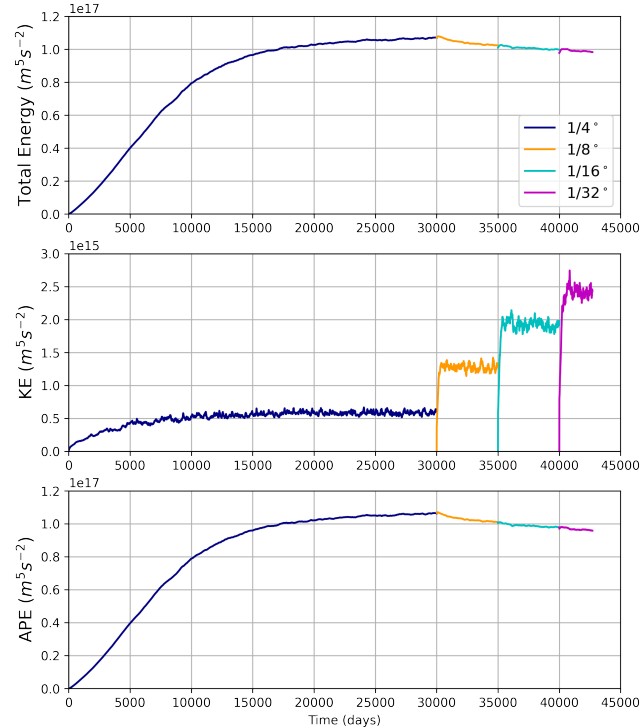

**Figure 3.** Time series of total energy (top panel), kinetic energy (middle panel), and available potential energy (bottom panel) as a function of time for all horizontal grids considered during spin-up and equilibration.

## 3 Spinup

The model is initialized from rest at $1/4°$ horizontal grid spacing, with initial conditions described in Section 2.2, and depicted in Fig. 1c. The $1/4°$ simulation is run to a quasi-steady state reached by about $3 \times 10^4$ days, in which the total kinetic energy

is no longer drifting. Next, the layer thicknesses are interpolated to the $1/8°$ horizontal grid, and the simulation is run again to a quasi-steady state for a few thousand days. Note that for expediency, velocities and transports are not interpolated but rather reset to zero. This step introduces a mild shock but the model quickly spins up mechanically, as seen in the recovery of kinetic energy levels in Fig. 3. This procedure is repeated for $1/16°$ and $1/32°$ horizontal grids until the simulation has reached convergence (see Section 5 for further description).

As the model horizontal grid is refined, the total kinetic energy (KE) increases at each transition to finer spacing (Fig. 3); this behavior is expected since the finer dynamical modes contain more kinetic energy. In addition, the available potential energy (APE) decreases at each transition to finer grids possibly because eddies are numerically better resolved and become more efficient at extracting APE. The total energy of the system drops at each transition in grid spacing, since APE dominates the total energy reservoir, but the drop is less than for APE due to the increase in KE.





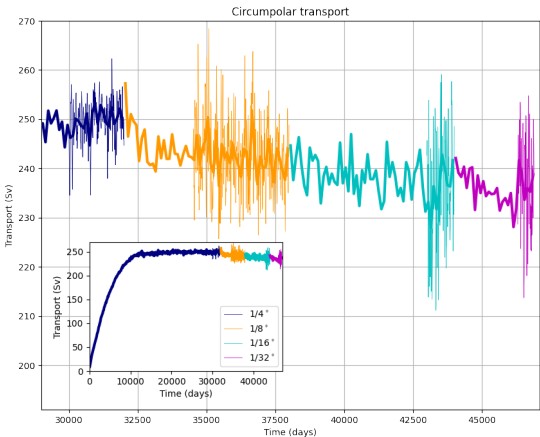

**Figure 4.** Time series of ACC transport, defined as the total (zonal) transport across longitude $0°E$ for each of the model horizontal grids during spin-up and equilibration. Thin lines are 5-day means (where available), thicker lines are 100-day means.

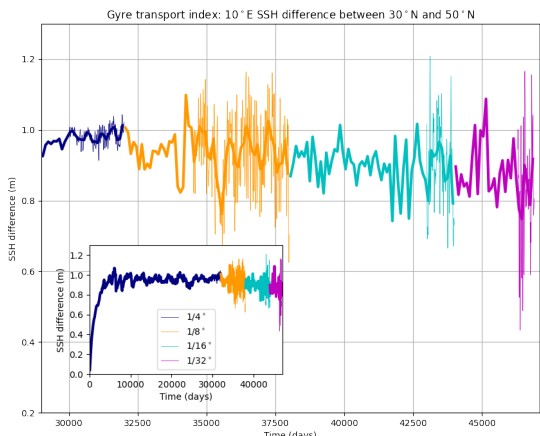

**Figure 5.** Time series of the gyre transport index, defined here as the difference in sea-surface height (m) at $10°$E between $50°$N and $30°$N for each of the model horizontal grids during spin-up and equilibration. Thin lines are 5-day means (where available), thicker lines are 100-day means.

The circumpolar transport spins up in $\sim 10^4$ days (Fig. 4) and fluctuates with no discernible drift for the remaining $2 \times 10^4$ days at 1/4°. There is a small reduction in circumpolar transport at each refinement in grid spacing, just as for APE. Since the baroclinic component of circumpolar transport is related to APE, arguments about the improved efficiency of eddies with refined grid spacing are relevant (Marshall and Radko, 2003).





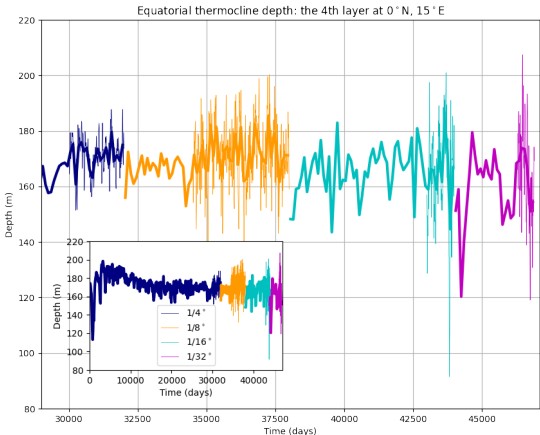

**Figure 6.** Time series of equatorial thermocline depth, defined here as the depth of the 4th layer, at longitude $15°E$ for each of the model horizontal grids during spin-up and equilibration. Thin lines are 5-day means (where available), thicker lines are 100-day means.

The northern hemisphere gyre transport, measured by the sea-surface height (SSH) difference between sub-tropical and sub-
polar regions, spins up in ~ 3000-5000 days in the 1/4° simulation and fluctuates without discernible drift for the remaining
$2.5 \times 10^4$ days (Fig. 5). This adjustment time is consistent with basin travel times for mid-latitude baroclinic Rossby waves
($\sim 2$ cm s$^{-1}$). The transition to finer grid spacing does lead to a small reduction in SSH difference but the changes, and any
drift, are well within the measured internal variability of the index.

The equatorial thermocline depth at 15°E adjusts on multiple timescales with large oscillations at the start of the 1/4°
simulation that damp out over ~4000 days (Fig. 6). A statistical equilibrium depth is reached on the order of $\sim 10^4$ days.
There is an adjustment at each transition in grid spacing but it is smaller than the dynamical noise.

During the spinup, the model exhibits multiple time-scales in the diagnostics described. The cost of the spinup to day
$4.2 \times 10^4$ (covering 1/4°, 1/8°, and 1/16°) is $\sim 17\%$ of the cost of the 2800 day segment at 1/32° resolution. The initializa-
tion procedure of consecutively allowing adjustment and then interpolating to finer grids is approximately $75\times$ cheaper than
spinning up the model solution entirely at 1/32°.

## 4   Mean Circulation and Mesoscale Turbulence

We illustrate the behaviour of the model in terms of key properties of the flow, with a focus on transport and energy reservoirs.
We compare the high-resolution NW2 with 1/32° horizontal grid, in which mesoscale eddies are explicitly resolved in the
majority of the domain, to lower resolution NW2 with horizontal grid of 1/4°, 1/8° and 1/16°. No mesoscale eddy closures are
used in any of the simulations beyond the Smagorinsky friction.



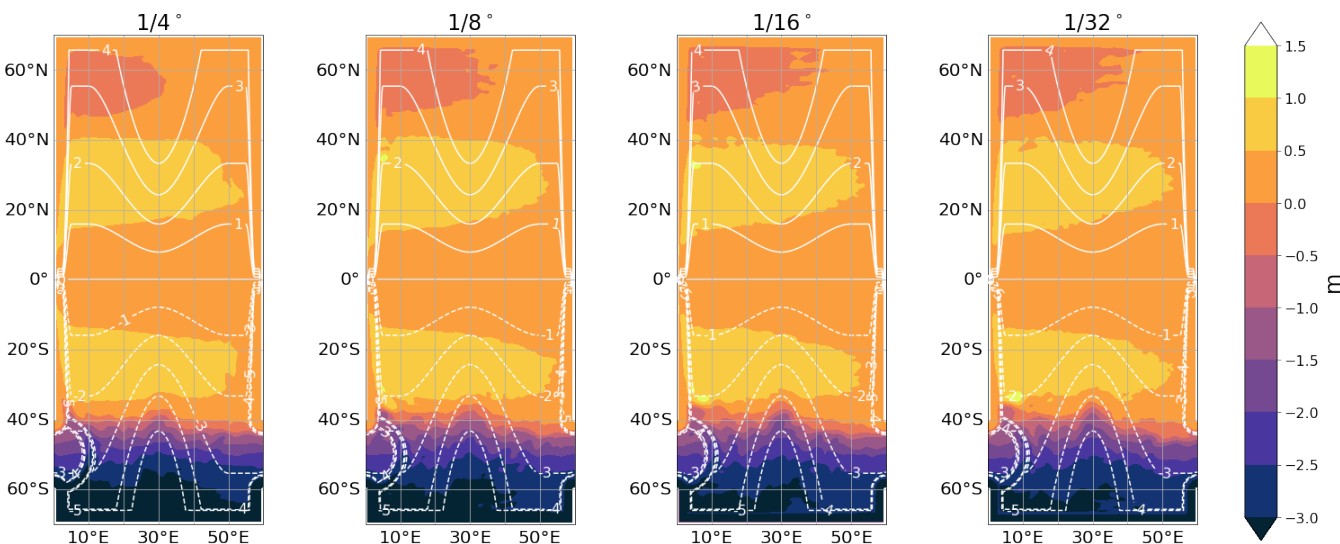

**Figure 7.** Time-mean sea surface height (SSH, m) averaged over the last 500 days of each simulation. White contours denote $f/H$ ($\times 10^{-8}$ m$^{-1}$ s$^{-1}$).

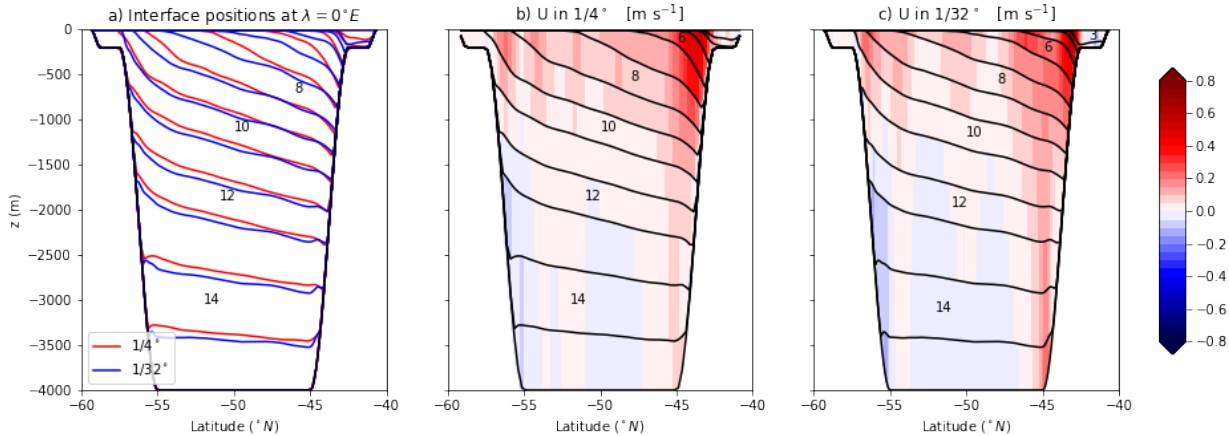

**Figure 8.** a) Mean hydrography across the Southern channel (Drake passage) for $1/4°$ and $1/32°$ models, depicted by time-mean position of interfaces (isopycnals). Time mean zonal flow (contour interval of 0.05 m/s), with interface positions for models at b) $1/4°$ and c) $1/32°$ horizontal grids, respectively. All results are averages over 500 days at longitude $0°E$, with eastward flow red and westward flow blue.

Most of the large scale features of the finest resolution configuration are recognizable in the coarsest resolution solution. The time-mean circulation is represented by subtropical gyres in both hemispheres, a subpolar gyre in the Northern Hemisphere (Fig. 7), a series of circumpolar jets in Southern Hemisphere reminiscent of the ACC (Figs. 8 and 9). The strength of the western boundary currents and their extensions (Fig. 5), and that of the circumpolar jet (Fig. 4) decrease slightly (proportionally) as




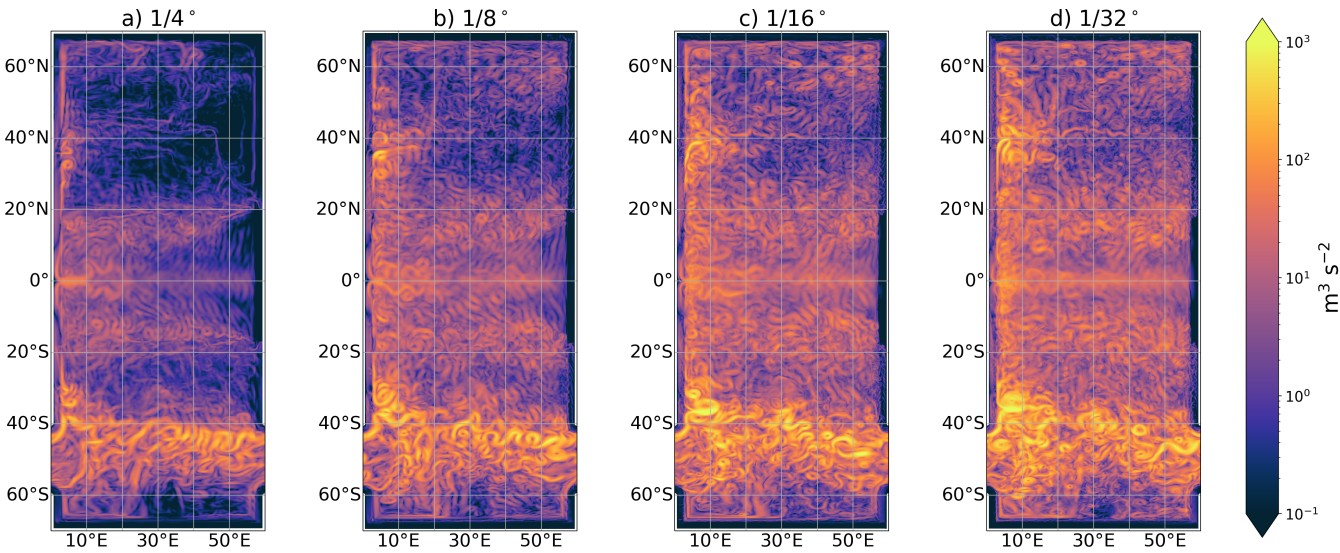

**Figure 9.** Snapshots of depth-integrated kinetic energy, computed as $\frac{1}{2}\sum_{k=1}^{N} h_k(u_k^2 + v_k^2)$.

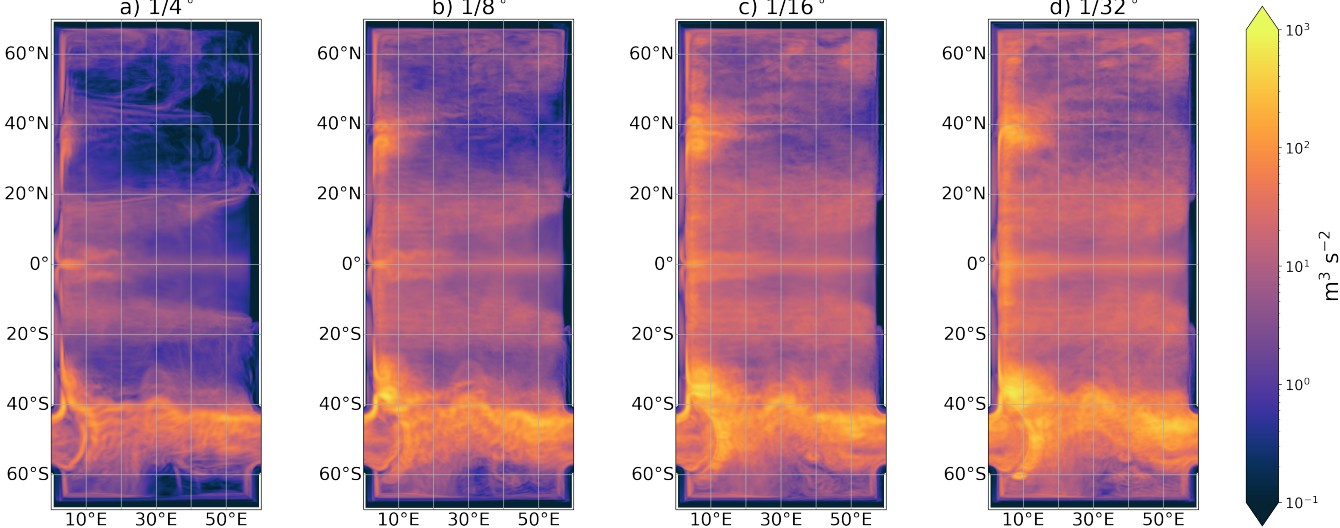

**Figure 10.** Depth-integrated kinetic energy as in Figure 9, but averaged over 500 days.

the grid spacing is refined. The thermocline depth along the equator also barely changes with resolution (Fig. 6). Overall, the character of the large scale circulation is relatively invariant with resolution even though the metrics of various features are converging with finer resolution.

Some details of the circulation that differ across resolution include emergent features. A major difference in upper ocean stratification is apparent in the Southern Ocean between the coarsest and finest resolutions. The interfaces in the fine resolution





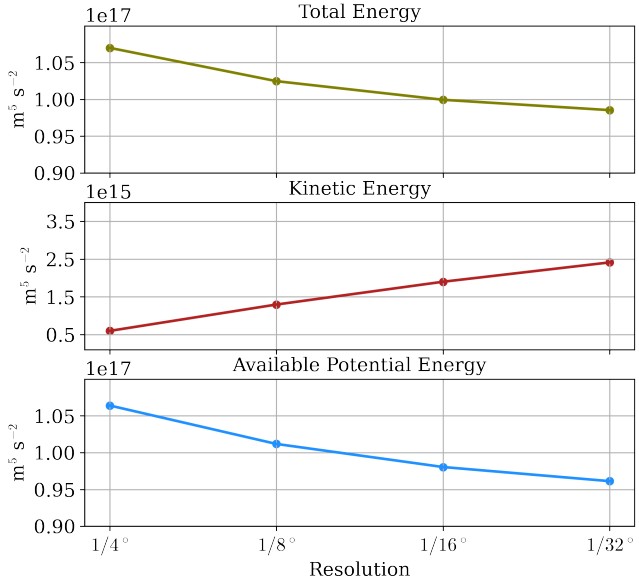

**Figure 11.** Equilibrium total energy, kinetic energy (KE), and available potential energy (APE) as a function of horizontal resolution, averaged over the last 500 days of each simulation. Note the different scale used for KE (middle).

simulation are less steep and the outcropping of interfaces move southward as a result of re-stratification by eddies (Fig. 8a). As the grid spacing is refined, the vertical extent of jets changes (Fig. 8). The time-mean zonal velocity shows either a change in the number of distinct jets or a migration of mean jet position. Most notable is the appearance of a deep westward circulation in the channel, below the depth of the blocking topography (the "Scotian" arc downstream of the passage).

Snapshots of depth-integrated KE reveal the eddying behaviour of the simulations (Fig. 9). For the coarsest grid (1/4°), the
flow permits mesoscale eddies, in particular at low latitudes where the deformation scale is largest. As the grid spacing is refined, the first deformation radius is better resolved, and mesoscale eddies become more ubiquitous and widespread. Note that a casual glance does not readily distinguish the 1/16° and 1/32° models.

The time-mean of depth integrated total KE (Fig. 10) becomes spatially smoother at finer grid spacing as eddies become increasingly ubiquitous. The same is true for sea surface height variance (not shown), a frequently used indicator of eddy
activity.

## 5  Convergence

Defining convergence for a turbulent cascade is challenging when using a dynamic viscosity because finer grid spacing will permit ever more variability on finer scales. For the purposes of mesoscale eddies, we are concerned with convergence as manifested by invariance of the large-scale properties as resolution changes, thus implying the upscale transfer of kinetic
energy is not changing so that details of the small-scales are not affecting the large-scale solution.



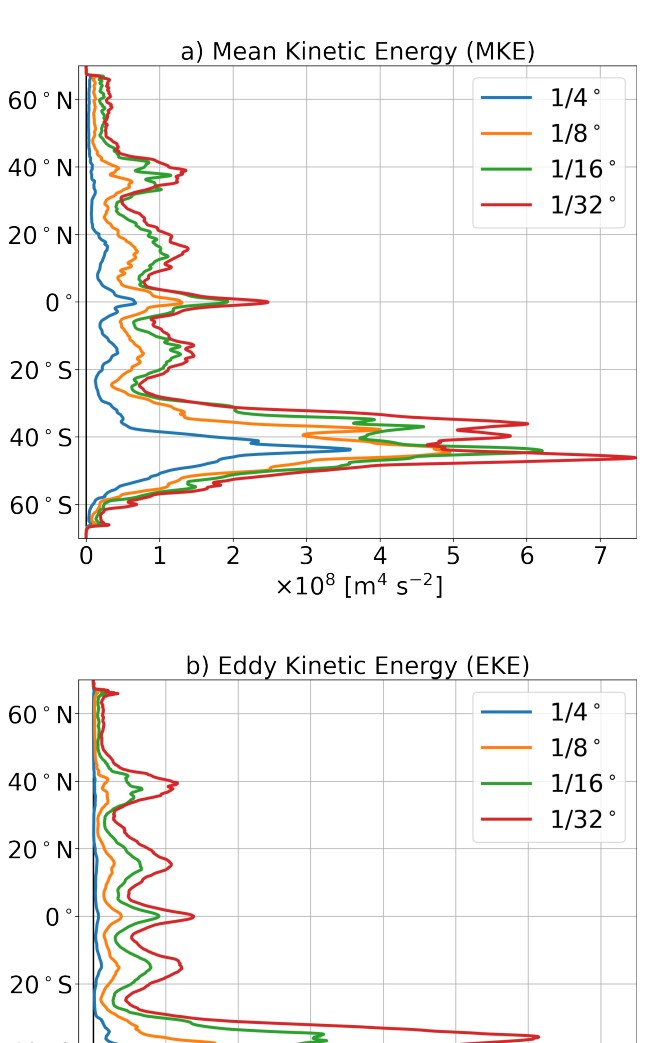

**Figure 12.** 500-day averaged, zonally- and depth-integrated a) mean kinetic energy (MKE, (8)) and b) eddy kinetic energy (EKE, (9)) for the four simulations in our model hierarchy. To compute MKE and EKE, we used a spatial filter with filter scale 1/4° in all four cases. Note that the x-axis in b) has been stretched by an order of magnitude compared to a).

The total mechanical energy of the system (APE + KE reservoirs), is dominated by the APE (Fig. 11 middle and bottom panels). APE is an integral metric of the system and the reservoir of energy that generates mesoscale eddies. In equilibrium, there is a balance of APE generation by winds (pumping/heaving the isopycnals) and the conversion of APE to mesoscale eddy energy, and the subsequent damping of mean and mesoscale energy by bottom drag, Smagorinsky friction, and vertical




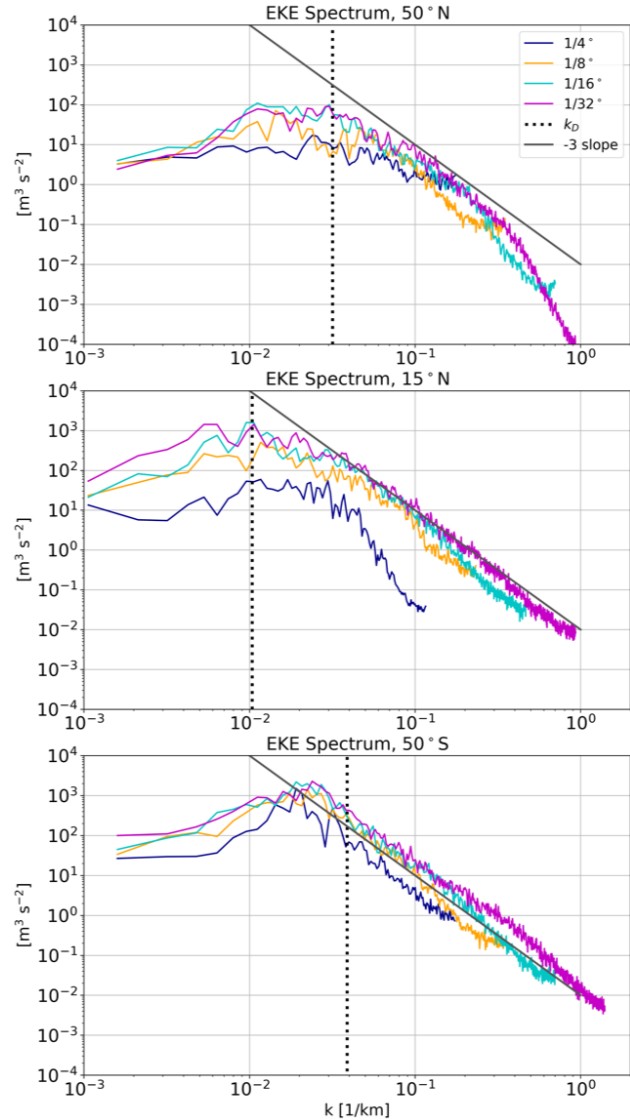

**Figure 13.** 100-day averaged spectra of surface EKE for all resolutions at three latitudes (top to bottom): $50°N$ (weak mean flow region), $15°N$ (mid-latitude gyre), and $50°S$ (ACC). Spectra are computed from the surface meridional eddy velocity fields, defined as deviations from a 100-day averaged meridional velocity (linear detrending and a Hann smoothing window are applied). The wavenumber $k_D$ corresponding to the deformation radius for the $1/32°$ simulation is shown as a dotted black line, and a $-3$ spectral slope is shown in grey.

viscosity. The equilibrium levels of energy reservoirs shown in Fig. 11 indicate a diminishing APE transition for each change in grid spacing, with the transition from $1/16°$ to $1/32°$ having the smallest change. We suggest that this behavior implies we are approaching convergence for resolving the mean APE-to-eddy energy pathway.





The kinetic energy increases less than linearly as grid spacing is refined (middle panel of Figs. 3 and 11); there is a five-fold increase in the kinetic energy reservoir between the 1/4° and the 1/32° simulations. To test (large-scale) convergence, Fig. 12

shows the split of 500-day averaged, zonally-and depth-integrated kinetic energy into scales larger than 1/4°, hereafter referred to as the "mean" kinetic energy (MKE, panel a), and scales smaller than 1/4°, hereafter referred to as the "eddy" kinetic energy (EKE, panel b). Specifically, we define

$$\text{MKE} = \frac{1}{2} \sum_{k=1}^{N} \bar{h}_k (\bar{u}_k^2 + \bar{v}_k^2) \,, \tag{8}$$

$$\text{EKE} = \frac{1}{2} \sum_{k=1}^{N} \overline{h_k (u_k^2 + v_k^2)} - \frac{1}{2} \sum_{k=1}^{N} \bar{h}_k (\bar{u}_k^2 + \bar{v}_k^2) \,. \tag{9}$$

The overbar denotes a spatial filter with filter scale of 1/4°. For the filter operation, we use the python package `gcm-filters` (Loose et al., 2022) with a Taper filter shape. Despite being close to a spectral truncation filter, the Taper filter still filters out some length scales larger than its filter scale of 1/4° (Grooms et al., 2021). Consequently, we observe a small amount of EKE even for the 1/4° simulation (blue line, Fig. 12b).

As the horizontal grid spacing is refined from 1/4° to 1/32°, we observe steadily growing eddy activity in all regions of the

domain (Fig. 12b). Kinetic energy at the large scales also intensifies with finer grid spacing (Fig. 12a), but the increase from 1/16° to 1/32° is smaller compared to the increase from 1/4° to 1/8° and from 1/8° to 1/16°. An exception is the energetic recirculation region north of Drake passage near 38°S, where MKE increases considerably as we change the grid spacing from 1/16° to 1/32°. We speculate that this increase is due to large-scale standing meanders that develop north of Drake passage due to finer resolved topography (Kong and Jansen, 2020; Barthel et al., 2017).

Overall, Fig. 12 suggests that as we refine the grid spacing in our model hierarchy from 1/4° to 1/32°, smaller scale eddies become more abundant and stronger, while the effect of eddies on the large scales is approaching an asymptotic limit.

The wavenumber spectra of KE at various latitudes (Fig. 13) show that at all latitudes, there is a clear gain in kinetic energy between the 1/4° and the 1/32° simulations, at all wavenumbers. As hinted by the snapshots of KE, convergence at low latitude is achieved faster than at high latitudes. The spectra generally suggest the large scales are more similar between 1/16° and

1/32° with spectral slopes and large scale magnitude apparently converged.

## 6 Conclusions

In this manuscript, we introduce an idealized ocean model configuration, NeverWorld 2 (NW2), that resolves mesoscale eddy dynamics in a pseudo-global context. NW2 is a stacked shallow water model using the MOM6 dynamical core (Adcroft et al., 2019), configured with a single cross-equatorial basin and a re-entering channel in the Southern Hemisphere with idealized

geometry. Because NW2 is strictly adiabatic, with no parameterizations in the vertical direction, the time-scales of adjustment are controlled entirely by dynamics rather than far slower thermodynamic processes. The paper serves as an introduction to the model and grid resolution hierarchy, and to the datasets for use by the community.



For the purposes of analyzing the role of mesoscale eddies, and deriving and testing parameterizations, we provide evidence that the finest grid spacing shown, 1/32°, is practically converged. The large scale metrics of APE and gyre transport, the latitudinal analysis of KE, and the spectral analysis of EKE, all suggest convergence of the largest scales when comparing solutions with the 1/16° and 1/32° grid spacings.

We have used the Smagorinsky dynamic biharmonic viscosity of Griffies and Hallberg (2000) for the momentum closure in these simulations, including for the finest grid spacing that defines our converged "truth". Previous work has shown a sensitivity of the details of the forward and inverse turbulent cascade to the form of dissipation (e.g., Smith and Vallis, 2002; Arbic and Flierl, 2004; Thompson and Young, 2007; Arbic et al., 2012; Pietarila Graham and Ringler, 2013; Soufflet et al., 2016; Pearson et al., 2017; Bachman et al., 2017; Treguier et al., 1997). In refining grid spacing here, we seek to converge on resolving the mechanism of interaction between the mesoscale eddies and the large scale circulation, by showing it to be diminishingly dependent on the details of dissipation near the bottom of the forward enstrophy cascade; this dissipation scale is more and more separated from the eddy production scale. This assumption could be tested by trying alternative closures and evaluating what tuning is necessary to give the same upscale energy flux. We could have used one of several closures proposed as scale-aware schemes to use in the eddy-permitting regime (e.g., Anstey and Zanna, 2017; Bachman et al., 2017, ,...). However, using them in the baseline configuration described here would hinder a fair evaluation of those schemes, which we plan to do so in the near future. Nevertheless, we acknowledge that the absolute magnitude of eddy energy, the span of the inverse cascade, and other metrics depend somewhat on the choice of viscous closure.

The model spatial-resolution hierarchy, from coarse- to eddy-rich, allows for a clean and extensive analysis of the dynamics and energetics of the flow as a function of horizontal grid spacing, which will be upcoming. The coarser grid configurations shown here can serve as a test bed for the evaluation of scale-aware mesoscale eddy parameterizations in the "grey" zone of eddy-permitting resolution. Even coarser, non-eddying configurations, not shown here because they need an eddy parameterization to look sensible, will be used to evaluate parameterizations of subgrid mesoscale eddies.

One virtue of the NW2 model is that the adiabatic limit isolates mesoscale eddies from other processes. However, there are strong interactions between mesoscale eddies and the surface mixed layer which have been recognized since early evaluations of mesoscale parameterizations (Danabasoglu et al., 1994). The dynamical core and algorithm used is the same as for a the full primitive-equation general circulation model (Adcroft et al., 2019; Griffies et al., 2020), so adding diabatic processes is relatively straightforward. The NW2 model can be modified and developed further to explore the large-scale overturning circulation (e.g., Wolfe and Cessi, 2009) or the connection with other parameterizations, for example, mixed layer and sub-mesoscale parameterizations.

*Code and data availability.* The MOM6 source code, NW2 configuration files and plotting/analysis scripts used in this article are available at https://doi.org/10.5281/zenodo.6462289. The entire NW2 data set, including initial conditions and restart files, will be publicly available via Open Storage Network.



*Author contributions.* GMM, NL, NB, JS, EY, CYC, AA, MFJ, RWH, LZ contributed to the development and analysis of NW2 configurations. GMM, NL, JS, EY, AA, LZ led the manuscript and SMG, BFK, MFJ, HK contributed to the manuscript.

*Competing interests.* The authors declare no competing interests.

*Acknowledgements.* GMM is supported by National Science Foundation (NSF) grant OCE 1912420. CYC and NB were supported by award NA19OAR4310365, and AA by award NA18OAR4320123, from the National Oceanic and Atmospheric Administration (NOAA), U.S.
Department of Commerce. LZ and EY were supported by NSF grant OCE 1912357 and NOAA CVP NA19OAR4310364. BFK is supported by NOAA NA19OAR4310366. NL was supported by NSF grant OCE 1912332. MFJ is supported by NSF grant OCE 1912163. JMS is supported by NSF grant OCE 1912302. SMG and RWH acknowledge support from the National Oceanic and Atmospheric Administration Geophysical Fluid Dynamics Laboratory. The statements, findings, conclusions, and recommendations are those of the author(s) and do not necessarily reflect the views of the National Oceanic and Atmospheric Administration, or the U.S. Department of Commerce. This
material is also based upon work supported by the National Center for Atmospheric Research (NCAR), which is a major facility sponsored by the NSF under Cooperative Agreement No. 1852977. Computing and data storage resources, including the Cheyenne supercomputer (doi:10.5065/D6RX99HX), were provided by the Computational and Information Systems Laboratory at NCAR, under NCAR/CISL Project Number UNYU0004 .





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
