# Peer review of "NeverWorld2: An idealized model hierarchy to investigate ocean mesoscale eddies across resolutions"

_EGUsphere, 2022_

## Author Response (AR1)

**Dear Dr. Wang,**

**Thank you for handling our manuscript. We are grateful to  Referee #1 and Dr. Takaya Uchida for taking the time to carefully read the manuscript and provide thoughtful comments.**

**Below are our detailed responses (in bold) to all the comments from RC1 and RC2.**

**Reply to RC1**

Summary

 This paper is supposed to serve as an introduction to a model configuration, grid resolution hierarchy, and to datasets for use by the community (lines 231–232). The model hierarchy (NeverWorld2) is reasonably well-described. It consists of a two-hemisphere basin with a southern reentrant channel realized using the adiabatic layered mode of MOM6. The idealized setup is similar to a number of previous iterations on the idea, although its use of adiabatic layer coordinates and analytically specified topography make the setup well suited for its intended use, which is to understand mesoscale eddy parameterizations and the impact of resolution on mesoscale eddy dynamics. To this end, the model hierarchy consists of four resolutions ranging in powers of two from 1/4º to 1/32º. The bulk of the manuscript is devoted to model description and comparison of the solutions at different resolutions.

The NeverWorld2 model hierarchy promises to be a useful tool for understanding mesoscale dynamics and I appreciate that the authors are releasing and documenting it. The manuscript is well-written and approachable overall, although there are some points that could be clarified and a some changes (discussed below) that would make the paper more useful and understandable. Perhaps the papers most significant shortcoming as an "introduction to datasets" is that the datasets themselves are not actually described, nor is a useable link or pointer to them provided to the reader. There is a statement in the "code and data availability" section that the dataset will be publicly available via Open Storage Network, but there's no indication of when or how they can be accessed. Information on the contents of the datasets beyond that they'll include "initial conditions and restart files" would also be useful. For example, it would be nice to know if the available data will also include mean fields, derived quantities (e.g., EKE, fluxes), and snapshots.

Other than the lack of discussion of datasets, most of my concerns relate to claims/demonstrations of convergence with resolution (see major comments below). I think the paper is quite publishable if these issues can be addressed. My sense is that the required revisions lie in the grey area between "major" and "minor"—I selected "major" but would not argue if the editor dropped it to "minor."

**We thank the reviewer for the positive and constructive feedback. The entire data set, including a detailed description of its contents, will be available via Geoscience Data**

**Exchange at https://doi.org/10.26024/f130-ev71. This DOI was not active when this comment was posted as some of the files were still being uploaded. We expect that this will be completed over the next two days and by then the DOI will be activated.**

Major comments

There are several attempts to argue that the 1/32º version is converged or approaching convergence, but a number are confusing or not convincing.

1. On the bottom of page 10 it is mentioned that the strength of the western boundary currents and their extensions decrease slightly with resolution. This seems rather unexpected: the transport of the WBC extensions is dominated by recirculation gyres that are largely eddy-driven. One expects that they would become stronger rather than weaker as resolution increases. Figures 9 and 10 indeed show substantial increases with resolution in the KE at the latitudes of the WBC extensions, so it's not clear where the claim that the strength of the extensions decreases with resolution comes from. The figure referenced in support (figure 5) merely shows SSH at 50 cm intervals, so it's impossible to tell what the WBCs are doing from this figure.

**We agree with the reviewer that the transport of the WBC extensions is dominated by eddies and should, therefore, become stronger as the grid spacing decreases. This is indeed in agreement with Figures 9 and 10. Figure 5 in the original manuscript shows a proxy for the surface gyre transport using the difference in sea surface height at 10 ºE between 50 ºN and 30 ºN. This metric can be misleading because it does not capture the barotropic effect of the eddies, which will increase with resolution. We have replaced Figure 5 from the original manuscript with Figure RC1a, which shows the time series of the difference in the barotropic transport stream function at 10 ºE between 50 ºN and 30 ºN. These points are outside the recirculation gyres and, therefore, this metric cannot be used to assess the effect of increasing grid spacing on the strength of the recirculation gyres.**

**We changed the third paragraph of Section 3 to:**

**"The difference in barotropic transport stream function between sub-tropical and sub-polar regions spins up in ~ 3000-4000 days in the 1/4 º simulation and fluctuates without discernible drift for the remaining $2.5 \times 10^4$ days (Fig. 5). This adjustment time is consistent with basin travel times for mid-latitude baroclinic Rossby waves (~ 2 cm/s). The transition to finer grid spacing does lead to an increase in the variability, which is a consequence of eddies becoming better resolved. However, the time-mean of this transport index does not change significantly with grid spacing, as expected via Sverdrup theory."**

[Figure]

**Figure RC1a: Time series of the barotropic transport at 10 ºE between 30 ºN and 50 ºN for each of the model horizontal grids during spin-up and equilibrium. Thin lines are 5-day means (where available), and thicker lines are 100-day means. The inset plot shows the entire time series, which starts from the 1/4º configuration at rest.**

**We have also replaced Figure 7 (we believe the referee mistakenly called this figure 5) with Figure RC1b, which shows the time-mean barotropic transport stream function averaged over the last 500 days of each simulation. We argue that this is a better metric for characterizing the mean circulation.**

**We modified the sentence at the bottom of page 10 to:**

**"Most of the large-scale features of the finest resolution configuration are recognizable in the coarsest resolution solution. The time-mean circulation (Fig. 7) is represented by subtropical gyres in both hemispheres, a subpolar gyre in the Northern Hemisphere, and a series of circumpolar jets in the Southern Hemisphere reminiscent of the ACC (Figs. 8 and 9). As the grid spacing is refined, the strength of the western boundary currents and their extensions seem to increase (Fig. 7), while the circumpolar jet (Fig. 4) decreases slightly (proportionally)."**

[Figure]

**Figure RC1b: Time-mean barotropic transport (Sv) averaged over the last 500 days of each simulation. Transport contours are shown every 10 Sv.**

2. Lines 201–202: The fact that the APE change from 1/16º to 1/32º is the smallest is used to suggest that the model is approaching convergence. However, a simple fit the APE curve shows that the APE is inversely proportional to the logarithm of the number of grid cells (in 1D). This means that the APE change after every doubling will be smaller than the previous doubling, but that the APE change will converge to zero very slowly with resolution.

**We thank the reviewer for the comment. Indeed, we were imprecise in these statements. To assess convergence we have now quantified the measurements. The model is approaching convergence in all energies, albeit slowly. We updated Figure 11 using a loglog scale, added a simple functional fit, and we indicate the limits reached at infinitesimal resolution (Figure RC1c). Details about these fits and their interpretation have been added to the manuscript.**

[Figure]

**Figure RC1c: The replacement for Figure 11 now shows a simple fit of A+B*(N-No)^p, and the asymptotic limit for infinitesimal resolution.**

3. The fact that the KE at scales greater than 1/4° appears to stabilize is a better metric of convergence than the APE change, although the fact that it's still increasing around 40ºS calls the convergence into questions. On the other hand, since the model hierarchy is designed to study mesoscale eddies, it would be more useful to know if the mesoscales(rather than the large scales), are converging. For this, a band-pass spatial filter with cutoffs several times larger and smaller than the local deformation radius would be more convincing.

**Thank you for this suggestion. We have updated Figure 12; it now shows the band-pass filtered kinetic energy. The convergence rate that we obtain seems very similar to what we saw for KE at scales greater than 1/4º (in the previous version): we**

**seem to approach convergence except for around 38 ºS, where standing meanders seem to play an important role.**

4. The fact that the spectra (figure 13) are collapsing onto each other is probably the most convincing indicator of convergence, such that it might be better to concentrate more on the spectra. A number of the aspects of the spectra are confusing, however.

**We have addressed the reviewer's comments regarding the spectra below. We added text to the caption of Figure 13 and an additional discussion of the spectra at the end of Section 5 to address these points in the manuscript.**

a) The spectra are supposed to be of EKE but are computed from the meridional velocity only. Unless the velocity is isotropic (unlikely in the presence of PV gradients), this would not necessarily give the EKE spectrum.

**We thank the reviewer for the comment. We had to make some practical choices to compute the spectra based on the configuration. The computation of the spectra proved to be challenging due to the relatively narrow domain size (see next comment). Given the small horizontal spatial extent of the domain, particularly at high latitudes, the large-scale flow is not well-resolved by the spectra. As such, we defined the 'eddy' components using deviations of the velocity field from a temporal average. Thus, to get a better sense of the eddy scales and mesoscale EKE spectra, we considered only the meridional eddy velocities that were less influenced by large-scale zonal current fluctuations. To illustrate this issue, please see the following two figures (Figures RC1d and RC1e). Figure RC1d contains spectra at 50 ºN computed using both zonal and meridional eddy velocities, while Figure RC1e is only using meridional eddy velocities. The latter proves more useful in assessing the EKE spectrum arising from mesoscale eddy features, and in identifying the energy-containing scale of the eddies. We do agree that the velocity is not isotropic and that the meridional spectrum does not give the full EKE spectrum; however, we still find this to be a useful way of visualizing the spectra given the constraints imposed by the model configuration. We have clarified some details of the spectra in the Figure 13 caption and in the end of Section 5.**

[Figure]

**Figure RC1d: EKE spectra computed using the meridional and zonal eddy velocities ('eddy' defined as a deviation from a temporal mean) at 50º North. Notice the large energy values at low wavenumbers (large scales) arising from fluctuations of the large-scale flow.**

[Figure]

**Figure RC1e: EKE spectra computed using only the meridional eddy velocities at 50º North. Notice the fall-off in energy towards large scales and the spectrum peaks at/slightly above the deformation radius, consistent with the dynamics of the inverse EKE cascade driven by mesoscale eddy activity.**

(b) The spectra are listed as being "at" several fixed latitudes. Are they zonal spectra or radial spectra?

**The spectra are zonal. We attempted to compute 2D radial spectra, but this presented challenges due to the domain geometry. The horizontal grid spacing in the zonal direction decreases with latitude, so when computing radial 2D spectra we first interpolate onto a fixed dx, dy grid ('fixed' in kilometers rather than degrees). Please see below (Figure RC1f) for an illustration. The spectra were affected by the interpolation. Also, the boxes over which the spectra were computed proved too small to capture the large-scale flow, particularly in the higher latitudes where the horizontal extent of the domain becomes small due to the Earth's curvature. So, we decided to present zonal spectra, which can be computed at each latitude without interpolation. It also allows us to make use of the full horizontal extent of the data at each latitude.**

[Figure]

**Figure RC1f: Shown is an example of non-interpolated surface KE density (left) vs. KE density that has been interpolated onto a regular grid with fixed dx=dy=5km (right). The data on the right was used to compute 2D radial spectra which proved less accurate than the approach of using zonal spectra on the native grid at each latitude.**

(c) The fact that the spectra at 15ºN and 50ºS follow inertial-range scaling all the way to the highest wavenumber is somewhat surprising. The forward cascade dissipates

enstrophy, so there should be at least a few wavenumbers in the enstrophy dissipation regime, but it seems like the cascade is dissipation-less all the way to the end.

**We find that the spectra at high wavenumbers are very sensitive to choices made in their computation, as expected. Specifically, we first cut off 2.5 º from the western and eastern boundary edges to remove boundary effects. We also apply linear detrending and a Hann window. Depending on which type of window is applied, how much of the boundary is cut off, and whether the data are interpolated onto a different grid (when we attempted to compute zonal spectra) the shape of the tail of the spectrum changes. For instance, below (Figure RC1g) is the same meridional spectrum that was shown in Figure RC1e but without detrending or a smoothing window. Due to the non-periodic domain and assumptions that go into the computation of the spectra, the high-wavenumber region is not necessarily behaving in a physical manner.**

[Figure]

**Figure RC1g: Same spectrum as shown in Figure RC1e, but without smoothing or windowing applied. Notice how the shape of the tail (high wavenumbers) changes.**

Minor comments

1. Line 87 and the rest of the page: The manuscript doesn't really provide an analysis of energetics, beyond tabulating the amount of energy in KE and APE in various ways. The conversion, transport, and generation terms in equations (6) and (7) are both standard and never used in the rest of the paper. At the moment they seem like filler and could be removed.

**Unfortunately, we already refer to these equations from other papers and we will therefore keep the equations. We added both more references to the equations and references to these papers in the manuscript.**

2. Lines 92–94: If the Coriolis terms do not actually vanish in a point-wise manner, they should be included in equation (6) if that equation is retained (but see above).

   **Correct. The non-zero Coriolis contribution has been included in the budget equations and explained that it should be zero but is not for numerical reasons.**

3. Lines 117–118: The idealized Scotia Arc is cute, but does it actually do anything? It may make the topography more "realistic" but there's nothing realistic about the topography in this model. The arc's appearance in an otherwise highly idealized model gives the impression that it's an important feature, but its impact is never discussed.

   **The Scotia Arc is necessary to remove momentum via form drag. This was now added to the manuscript.**

4. Figures 4 & 5: The insets should be described in the captions.

   **A description of the insets has been added in the captions of Figures 4 & 5.**

5. Figure 7: The contour spacing for SSH (50 cm) is too coarse to see detail of anything other than the circumpolar current. The entirety of each gyre is represented by a single contour, which doesn't make it a very useful diagnostic. Further, the $f/H$ contours aren't mentioned in the text and don't seem to add anything (the topography is shown in figure 1a) except to clutter up the figure.

   **We have replaced Figure 7 with Figure RC1b. See response to major comment #1.**

6. Line 178: How are "emergent features" defined? This phrase appears to include features such as southern ocean stratification and the vertical extent of jets, but not features such as tropical thermocline depth and the strength of the WBCs and circumpolar jet.

   **Yes, good point. We edited these sentences and no longer refer to "emergent features" which are not well defined nor obvious in this series of experiments.**

7. Lines 205–210: Given that time mean energy has already been discussed, switching the definition of "mean" to mean "scales larger than 1/4°" is confusing terminology. Something like "large scale energy" and "small scale energy" would be better. ("Mesoscale energy" implemented using a band-pass filter would be even better.)

   **Thanks for pointing this out. We changed the terminology to "mesoscale energy", following the suggested band-pass filter approach.**

8. Line 228: "Basin-scale" would be more accurate/clear than "pseudo-global".

**Changed.**

Minor issues (grammar, typos, etc)

1. Lines 30–31: The last sentence of this paragraph makes it seem like there's going to be a description of "recent mesoscale parameterizations … with novel momentum closures" but the paragraph simply ends after this sentence.

   **Addressed.**

2. Line 44: "The broad configuration is similar" should be "The configuration is broadly similar" (unless there's some distinction between a broad configuration and a narrow configuration I'm missing).

   **Changed.**

3. Line 68: "we here write" → "here we write"

   **Changed.**

4. Line 87: "and in subsequent papers." What will happen in subsequent papers? It seems like there should be more after "papers".

   **Thank you. We have added references to the relevant papers using the NW2 dataset.**

5. Lines 176–177: "features are converging" should be "features converge" to match the tense of the rest of the sentence.

   **Changed.**

6. Line 180: "the outcropping of interfaces" â "the interface outcrops"

   **Changed.**

7. Line 246: There appears to be an unresolved reference following Bachman et al., 2017.

   **Fixed.**

**Reply to RC2**

Marques et al. introduce a new purely adiabatic primitive equation model which is computationally cheap and easy to run. As they note, a cheap and versatile model to test mesoscale eddy parametrizations has indeed long been a needed tool for the ocean modeling community and their configuration would be a great resource for the community. NeverWorld2 (NW2) being part of the MOM6 module also provides

confidence in the stability of their model. The manuscript is well written and I only provide minor points listed below.

- Some discussion regarding how computationally cheap NW2 is compared to a non-adiabatic, isopycnal primitive equation model where the equation of state for density is linear (e.g. the density linearly depending only on temperature without salinity) would be nice to have. While I understand the adiabatic nature of NW2 allows the user to focus on the dynamics and isolate mesoscale processes, a non-adiabatic isopycnal model is closer to reality, also allowing for a surface mixed layer.

**We appreciate the reviewer's comment on this point. We have made the choice of using an adiabatic model to indeed isolate the effects of mesoscale eddies. Unfortunately, we cannot give the exact cost of running NW2 in the diabatic mode because this configuration does not exist. The main reason for using an adiabatic configuration is that the model achieves an equilibrated state significantly faster than with a diabatic mode. It would take 1000's years for the deep ocean to equilibrate in a diabatic setup, while we were able to achieve this in 10's years for the ¼ degree configuration. Thus, roughly, the cost of NW2 is 100x less than that of a diabatic run. In addition, running the model in the diabatic mode would require an increased number of vertical levels to represent the surface boundary layer, which would make the model more computationally expensive.**

- Figure 13: Is any tapering applied prior to taking the Fourier transform to make the data periodic?

**Thank you for the question. Yes, when computing the spectra we use the XRFT Python package (https://xrft.readthedocs.io/en/latest/) with a Hann window to taper the data. Since the data are not periodic and influenced by boundary effects, we also cut off 2.5 degrees from the Western and Eastern boundaries before computing the spectra. We added text to the caption of Figure 13 and additional discussion of the spectra at the end of Section 5 to address these points.**

---

## Author Response (AR2)

Dear Dr. Wang,

The updated manuscript includes the suggestions provided for improving the figures. We increased the font size and also enlarged the figures. We also addressed the minor suggestion for line 179 and included the country names in the addresses of all authors.

Thank you,

Gustavo